# Optimal Selection for Redox Couples and Enhanced Performance through Magnetic Nanofluid Electrolyte in Solar Flow Batteries

Zixing Gu [1,†], Ping Lu [1,†], Zihan Zhang [1], Qiang Ma [1], Huaneng Su [1] and Qian Xu [1,2,*]

1   Institute for Energy Research, Jiangsu University, Zhenjiang 212013, China;
    2222206025@stmail.ujs.edu.cn (Z.G.); 2111906004@stmail.ujs.edu.cn (P.L.);
    2212106023@stmail.ujs.edu.cn (Z.Z.); maqiang@ujs.edu.cn (Q.M.); suhuaneng@ujs.edu.cn (H.S.)
2   Jiangsu Provincial Engineering Research Center of Key Components for New Energy Vehicle,
    Wuxi Vocational Institute of Commerce, Wuxi 214153, China
*   Correspondence: xuqian@ujs.edu.cn; Tel.: +86-511-88799500
†   These authors contributed equally to this work.

**Abstract:** The limited photoelectric conversion efficiency poses one of the critical constraints on commercializing solar flow batteries (SFBs). This study compares the chemical and photoelectrochemical properties of three commonly used redox couples. Additionally, magnetic $Fe_3O_4$ nanoparticles, for the first time, are introduced to optimize the electrolyte, and they are compared with the original electrolyte. Across different redox couples, the variations in semiconductor flat-band potentials and carrier concentrations result in changes in photoelectric current density. Notably, $FeCl_2/FeCl_3$ redox coupled with $TiO_2$ photoelectrodes exhibits the highest photoelectric current density, reaching 75.7 $\mu A\ cm^{-2}$. However, the trade-off of this electrolyte, i.e., providing high photocurrent while being unable to supply sufficient open-circuit voltage, imposes limitations on the practical application of SFBs. Alternatively, for TEMPO and 4-OH-TEMPO electrolytes, which can provide a higher open-circuit voltage, the electrochemical activity is enhanced, and the solution ohmic resistance is reduced by introducing magnetic nanoparticles to form a magnetic nanofluid. As a result, the photoanode's photocurrent density increases by 36.6% and 17.0%, respectively, in the two electrolytes. The work reported here effectively addresses the current issue of low photocurrent density in SFBs and presents new optimization strategies for SFBs.

**Keywords:** solar flow batteries; redox couples; magnetic nanofluid; photoelectric conversion efficiency; photocurrent density





## 1. Introduction

Against the backdrop of a series of environmental issues triggered by traditional energy sources, such as climate change, acid rain, and air pollution, humanity's energy landscape is undergoing a transition toward green and low-carbon development. Approximately $1.8 \times 10^{11}$ MW of solar energy intercepted by the Earth makes solar energy the most abundant and cleanest among all renewable energy sources to date [1]. The main drawback of solar energy is its intermittent and unstable nature, as it cannot provide power at night and exhibits strong fluctuations under partially cloudy conditions [2]. Photovoltaic cells and photoelectrochemical cells (PECs) represent the two most common forms of solar energy utilization [3]. It is essential to store the converted electrical energy to overcome the drawbacks of solar energy intermittency and instability and ensure its availability. Flow batteries have attracted significant attention among various energy storage technologies due to their advantages, including low emissions, low cost, flexible capacity design, and safety. The traditional solar-flow battery energy storage system involves connecting a photovoltaic solar panel array to a flow battery via conductors, along with regulating maximum power point tracking (MPPT) devices and various converter devices [4], making the entire system relatively complex.

Recently, the integration of photoelectrochemical cells and redox flow batteries in an SFB has gained special attention due to its simple and compact structure. Compared to traditional storage structures, it eliminates the need for expensive MPPT and various conversion devices, making it more suitable for applications such as household solar systems in remote or rural areas, microgrids on islands and offshore oil platforms, or small off-grid systems [5,6]. Solar flow batteries face several challenges that impede their development: (i) extremely low working current (<0.1 mA/cm$^2$) and efficiency (<0.4%); (ii) relatively low energy density of the electrolyte and a biased charge–discharge state of charge (SOC) (<80%); (iii) incomplete integration design of the battery system; and (iv) the need for improvement in battery lifespan and stability [7,8]. Among these, the significantly low working current and photoconversion efficiency have been persistent primary obstacles to their development. For SFBs, the photoelectrochemical behavior at the semiconductor/solution interface is the most critical aspect of the system and currently has the most significant impact on system performance. As illustrated in Figure 1, taking the common n-type semiconductor as an example, when the semiconductor comes into contact with the electrolyte, the Fermi level at the interface is influenced by the redox couples in the solution, resulting in the bending of the semiconductor's interface energy bands. When the Fermi level of the photoanode is higher than the potential of the redox couples in the solution, electrons flow from the semiconductor to the solution, leading to the formation of a space charge region with a relatively high positive charge concentration on the semiconductor side of the interface. The substitution of different redox couples can influence the ratio of the oxidation state to the reduction state on the surface of the photoanode, thereby affecting the photoanode's charge density and electron states. These changes impact various interface properties of the photoanode, such as the flat-band potential and carrier concentration. Therefore, the selection of redox couples can be used to regulate the distribution of surface traps and unoccupied electron states, optimizing the efficiency of photoelectronic separation and subsequently improving the overall photoconversion efficiency [9–11].

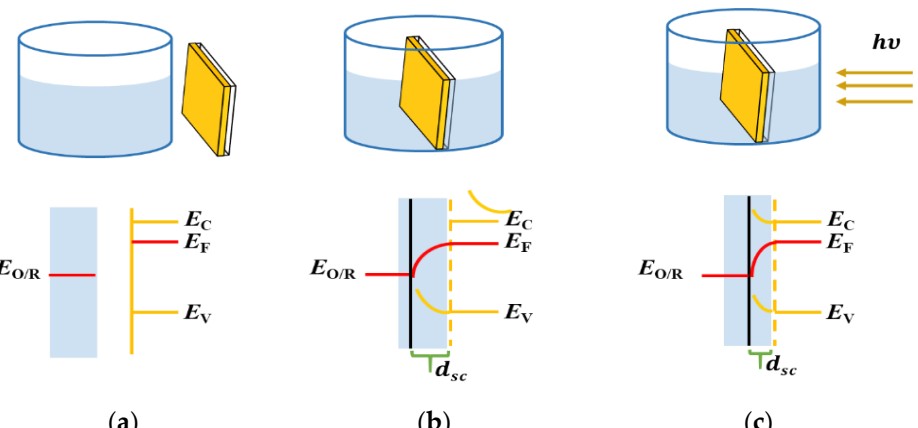

**Figure 1.** The interface between the photoanode and the electrolyte: (**a**) no contact; (**b**) contact; (**c**) light illumination.

Most research efforts, however, focus predominantly on the solid component within the solid–liquid interface. In research, a single electrolyte is usually used as a quantitative measure to optimize the design of the photoelectrode. There is a limited amount of research comparing the redox couples in the electrolyte on the photoanode side and their interface. In fact, the choice and optimization of the electrolyte are equally critical. On the one hand, it can impact photoconversion efficiency, and on the other hand, it plays a crucial role in determining the battery's capacity density, open-circuit voltage, stability, and lifetime. In the selection of redox couples, various well-performing redox flow batteries (RFBs), such as iron-based RFBs [12–14] and vanadium redox flow batteries (VRFBs) [15],

have been integrated with different photoelectrodes. Emerging redox couples, including 2,2,6,6-tetramethylpiperidine-1-oxyl (TEMPO) [16–18] and its derivative 4-OH-TEMPO [19], have also garnered widespread attention in solar flow batteries [20–22]. Regarding the choice of solvents, non-aqueous deep eutectic solvent (DES) electrolytes address the issue of narrow electrochemical windows in aqueous systems, which can lead to the impact of oxygen/hydrogen evolution side reactions on open-circuit voltage and stability. Moreover, unlike organic solvents, they do not pose toxicity concerns [23,24]. Therefore, our study opted for a DES obtained by combining choline chloride and ethylene glycol as the electrolyte solvent. $FeCl_2$, TEMPO, and 4-OH-TEMPO were preliminarily selected as the anodic electrolytes due to their favorable performance in the DES.

Additionally, numerous studies have indicated the crucial role of introducing additives in enhancing electrolyte electrochemical reaction kinetics and stability. Wang et al. investigated the use of $In^{3+}$ as an additive to improve the stability and performance of iron-chromium flow batteries (ICFB) [25]. Cao et al. studied the influence of additives in VRFB on battery performance, electrochemical kinetics, and energy density [26]. Kim et al. introduced carbon-based nanoparticles into the electrolyte of a vanadium redox flow battery, demonstrating that the nanofluid electrolyte exhibited superior oxidation/reduction kinetics and mass transport compared to the original electrolyte [27–30]. There have been no attempts in SFB research to enhance battery performance by introducing additives into the electrolyte. Magnetic nanofluids have shown great potential in various applications related to heat transfer, including microelectronics, fuel cells, and engine cooling/vehicle thermal management, owing to their enhanced thermal conductivity and convective heat transfer coefficient compared to base fluids. In recent years, recyclable magnetic nanoparticles have gradually demonstrated significant potential in electrochemistry [31,32]. Hence, preparing nanofluid with magnetic nanoparticles as additives to enhance the photoelectrochemical conversion efficiency of solar flow batteries (SFBs) holds significant research potential.

This paper focuses on comparing the chemical properties of several analytes in SFBs and their photoelectrochemical performance on the anodic side. We observed that the electrolyte could not generate a strong photocurrent density while providing a high open-circuit voltage. To address this issue, we initially attempted to enhance the electrolyte's physicochemical properties and optimize the photoelectrochemical performance by introducing magnetic nanoparticles. By harnessing the inherent Brownian motion of nanoparticles, we aim to enhance the mass transfer capability of the electrolyte in solar flow batteries. The study reveals that these magnetic nanofluid electrolytes increase the photoelectric current density of the photoanode, enhancing the efficiency and usability of the battery. This advancement contributes to the development and application of SFBs.

## 2. Materials and Methods

### 2.1. Chemicals

2,2,6,6-tetramethylpiperidinyl-1-oxide (TEMPO), 4-hydroxy-2,2,6,6-tetramethyl-piperidinooxy (4-OH-TEMPO), ferrous chloride tetrahydrate ($FeCl_2 \cdot 4H_2O$), vanadium chloride ($VCl_3$), tetrabutyl titanate ($C_{16}H_{36}O_4Ti$), choline chloride ($C_5H_{14}ClNO$), absolute ethanol ($CH_3CH_2OH$), polyethylene glycol ($HO(CH_2CH_2O)_nH$), iron trichloride hexahydrate ($FeCl_3 \cdot 6H_2O$), deionized water ($H_2O$), and FTO glass were used in this study. All chemicals were analytical grade and used without further purification.

### 2.2. Preparation of Magnetic $Fe_3O_4$ Nanofluid Electrolyte

TEMPO, 4-OH-TEMPO, and $FeCl_2 \cdot 4H_2O$ were dissolved in deep eutectic solvent (DES) with a molar ratio of choline chloride to ethylene glycol of 1:2. The molar concentration was 0.1 mol/L, serving as the original electrolyte on the photocathode side. The cathode electrolyte was 0.1 M $VCl_3$, and it was also dissolved in the same solvent as a quantitative reference.

$Fe_3O_4$ nanoparticles were synthesized using a chemical co-deposition method [33,34]. A mixture of 40 mL ethanol and 8 mL water was combined, which was followed by the

addition of 0.99 g $FeCl_2 \cdot 4H_2O$ and 2.70 g $FeCl_3 \cdot 6H_2O$. Ammonia solution with a mass fraction of 25% was slowly added until the pH reached approximately 10. The reaction was carried out at a constant temperature of 45 °C with stirring for 15 min, which was followed by maturation at 75 °C for an additional 15 min. It is important to note that nitrogen gas was purged during the experiment to prevent oxidation of the formed magnetic $Fe_3O_4$. The solution was cooled to room temperature, and under the influence of a magnetic field, the nanoparticles were separated, washed with deionized water and anhydrous ethanol, and vacuum-dried for 12 h before being ground for further use.

Magnetic $Fe_3O_4$ nanoparticles were added to the prepared DES electrolyte at mass percentages of 0.01 wt%, 0.025 wt%, and 0.05 wt%, which were followed by constant stirring at a controlled temperature for 2 h to prepare the magnetic nanofluid electrolyte.

### 2.3. Preparation of Photoanode

A nanorod array of $TiO_2$ was prepared on FTO glass using a hydrothermal method consisting of the following steps as the photoanode [35,36]. Mix 12 mL of concentrated hydrochloric acid (36~38%) with an equal volume of ultrapure water and stir for 5 min. Then, add 0.4 mL of titanium butoxide and continue stirring for 5 min. Once the mixture of the precursor solution becomes clear, transfer it to a 100 mL stainless steel autoclave. Place the FTO glass at an angle inside the autoclave with the conductive side facing down. Subject it to hydrothermal treatment at 150 °C for 6 h. After the autoclave cools to room temperature, remove the FTO glass, rinse it with deionized water, and dry it. A white thin film, $TiO_2$, will be observed attached to the surface. Finally, place it in an alumina crucible and anneal at 450 °C in air for 1 h.

### 2.4. Electrochemical Measurements

The electrochemical performance of the electrolyte/photoanode interface was tested using a Dutch Ivium electrochemical workstation. In the experimental setup, a three-electrode system was employed with either a glassy carbon electrode (for the flow cell half-cell testing mode) or a 1 $cm^2$ $TiO_2$ photoanode (for the solar flow battery half-cell testing mode) as the working electrode in the electrolyte. A platinum electrode served as the counter electrode, and a saturated calomel electrode (SCE) was used as the reference electrode. For the flow cell half-cell mode, cyclic voltammetry and electrochemical impedance spectroscopy were conducted. In the solar flow battery half-cell mode, linear sweep voltammetry, Mott–Schottky curves, and transient current scans were performed.

### 2.5. Characterization of Materials

X-ray diffraction (XRD) was performed using the D8 Advance instrument from Bruker, Germany, with a Cu Kα source (λ = 1.5418 Å). The scanning speed was set at 7° $min^{-1}$, and the measurement angle range was 20° to 80°. Qualitative analysis was conducted by comparing the characteristic peaks of the material with those on standard diffraction cards. The viscosity and conductivity of the electrolyte have a significant impact on the transport of active materials within the battery, making them crucial physical parameters. These properties were tested using a DV-2+PRO digital viscometer (Shanghai Nirun, Shanghai, China) and DDS-307A digital conductivity meter (Shanghai Zhiguang, Shanghai, China). Furthermore, the morphological dimensions of the $Fe_3O_4$ nanoparticles were examined using Gemini 300 scanning electron microscopy (Zeiss, Oberkochen, Germany) at an operating voltage of 10 kV. Ultraviolet–visible (UV-Vis) measurements were conducted on the original electrolyte and nanofluid electrolyte diluted 50 times using a UV2600 UV-Vis spectrophotometer (Shimadzu, Kyoto, Japan) within the wavelength range of 200–500 nm.

## 3. Results and Discussion

### 3.1. Electrochemical and Photoelectrochemical Performance of Different Redox Couples

The electrochemical performance of different redox couples was characterized through cyclic voltammetry (CV) and electrochemical impedance spectroscopy (EIS). As shown

in Figure 2a, all three redox couples exhibit distinct redox peaks, indicating their rapid redox kinetics, which is a prerequisite for them to serve as electrolytes. Due to the different redox potentials of the three electrolytes, the open-circuit voltage generated by the cell varies when $VCl_3$ is used as the cathodic electrolyte. When 4-OH-TEMPO is the anodic electrolyte, the highest open-circuit voltage is 1.17 V. Furthermore, the ratio of the absolute value of the oxidation peak current to the absolute value of the reduction peak current, known as the peak current ratio, is greater than 1 for all three electrolytes. This suggests fast oxidation reaction rates, indicating that electrons readily flow out of the substances, making them suitable for use as electrolytes on the photoanode side. The peak current ratio closest to 1 is the $FeCl_2$ electrolyte, indicating its optimal reversibility. The peak currents of the TEMPO and 4-OH-TEMPO electrolytes are slightly higher than those of the $FeCl_2$ electrolyte, attributed to the faster electron transfer rate, resulting in a higher current generation.

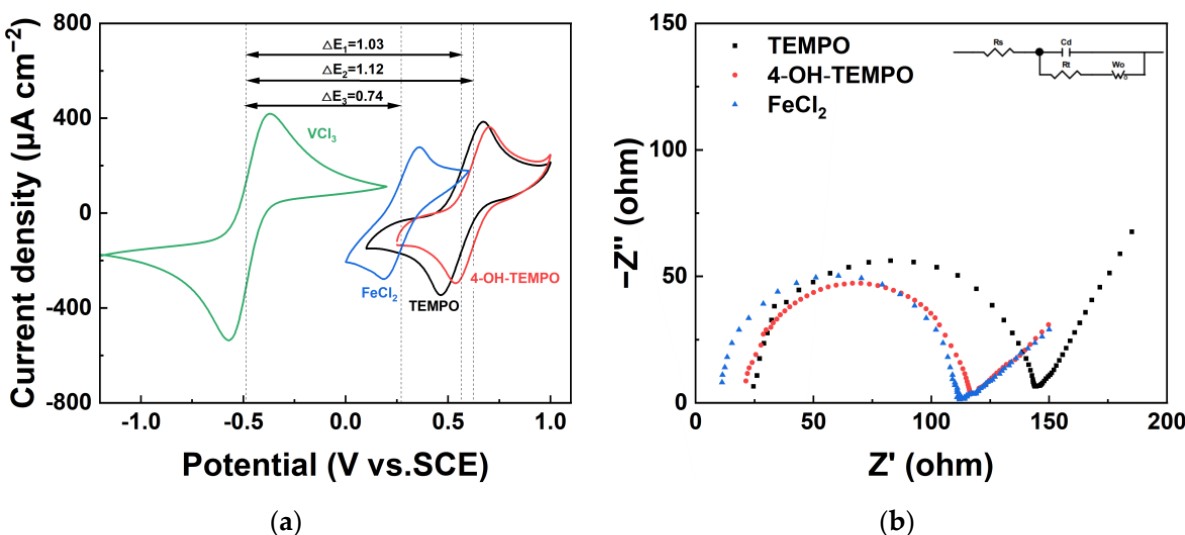

**(a)** **(b)**

**Figure 2.** (**a**) Instantaneous current plots; (**b**) electrochemical impedance spectroscopy of different electrolytes.

The impedance spectra in Figure 2b consist of a semicircle in the high-frequency region and a straight line in the low-frequency region, indicating a mixed control of the electrochemical reaction and diffusion steps in the redox reaction process. The semicircle reflects the ohmic resistance ($R_s$) and charge transfer resistance ($R_t$) in the transfer reaction at the electrode–electrolyte interface. Based on the equivalent circuit illustrated in the figure, fitting was conducted using Zview-4 software, and the obtained data are presented in Table 1. Analysis indicates that the solution resistance of $FeCl_2$ is the smallest, with a fitted impedance spectrum yielding an ohmic resistance of only 9.75 ohm, which is less than half of that in TEMPO and 4-OH-TEMPO solutions. Among the three solutions, 4-OH-TEMPO exhibits the lowest charge transfer resistance, measuring 94.66 ohms, implying an accelerated charge transfer of the redox pair in the solution, leading to an increased electrochemical reaction rate.

**Table 1.** The parameters resulting from fitting the impedance plots with the equivalent circuit model in Figure 2b.

|  | $R_s/\Omega$ | $R_t/\Omega$ | $W_o/\Omega$ |
|---|---|---|---|
| TEMPO | 24.50 | 116.2 | 24.1 |
| 4-OH-TEMPO | 20.75 | 94.66 | 214.4 |
| $FeCl_2$ | 9.75 | 100.8 | 179.0 |

We analyzed the photoelectrochemical performance between different redox couples and the $TiO_2$ photoanode by testing the instantaneous current curves under illumination and Mott–Schottky plots. After exposure to light, Figure 3a illustrates that the photocurrent density stabilizes at 75.7 µA cm$^{-2}$ for $FeCl_2$, 18.5 µA cm$^{-2}$ for TEMPO, and 39.3 µA cm$^{-2}$ for 4-OH-TEMPO. The intersection of the straight portion of the Mott–Schottky plot with the x-axis was used to determine the flat-band potential of the semiconductor electrode, while the slope of the curve allowed for the calculation of carrier concentration. From the analysis of Figure 3b, it is observed that the flat-band potentials of the semiconductor in the three electrolytes differ, with $FeCl_2$ being the most negative at $-0.46$ V. In contrast, the flat-band potentials under TEMPO and 4-OH-TEMPO are relatively close, around $-0.32$ V. The carrier concentration at the photoanode in $FeCl_2$ is approximately $5.7 \times 10^{17}$, which is about twice that in the other two electrolytes.

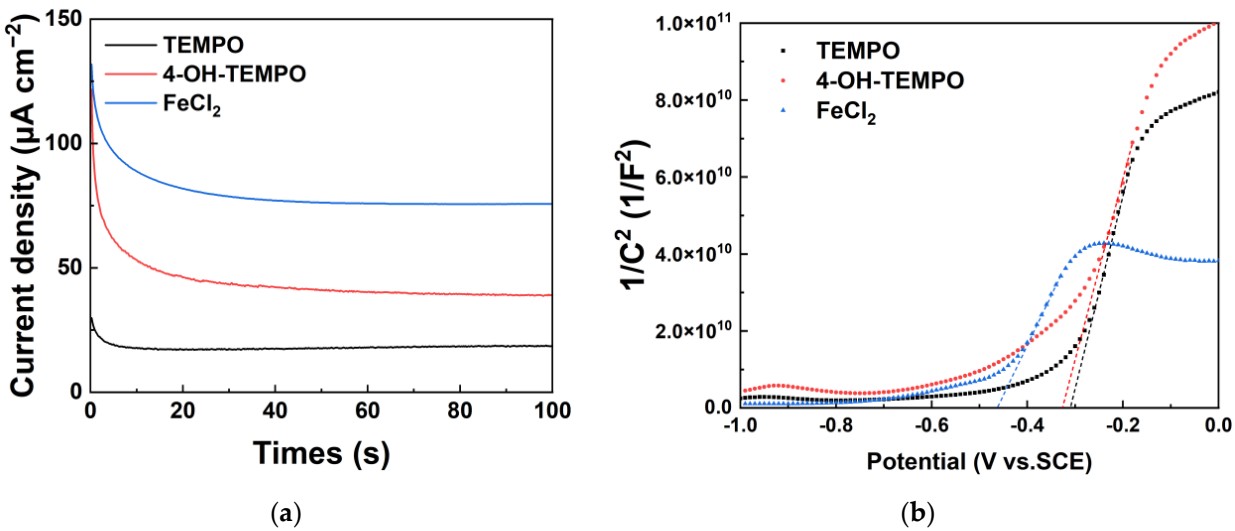

**Figure 3.** (**a**) Instantaneous current plots; (**b**) Mott–Schottky plots of $TiO_2$ photoelectrode with different electrolytes.

Three redox couples exhibit variations in electrochemical properties such as redox potential, electrocatalytic activity, and ohmic resistance. The matching of $TiO_2$ photoelectrodes with different electrolytes reveals diverse photoelectrochemical performances. Analysis indicates a correlation between the shift in $TiO_2$ flat-band potential and the occurrence of oxidation reactions. The disparity in the electrolytes' redox potentials directly influences the photoelectrode's flat-band potential, emerging as a primary factor contributing to the differences in photoelectrochemical performance. The oxidation–reduction potential of $FeCl_2$ is relatively low, resulting in a more negative flat-band potential formed between $TiO_2$ and the $FeCl_2$ solution. This leads to a relative lowering of the semiconductor's valence band top level, an increase in the conduction band bottom level, and a narrowing of the bandgap width. In this scenario, the energy barrier between the surface reactants and the conduction band increases, enhancing the activity of surface oxidation reactions promoting charge separation and consumption reactions, thereby enhancing the reactivity of surface reactants. Additionally, the lower ohmic resistance and charge transfer resistance are also reasons for the higher photocurrent in $FeCl_2$. In the presence of TEMPO and 4-OH-TEMPO electrolytes, where the redox potentials are closely aligned, there is a notable discrepancy in the photocurrent density generated by the photoelectrode under illumination. The analysis attributes the lower photoconversion efficiency to elevated ohmic resistance and charge transfer resistance within the TEMPO solution. Consequently, the resistance in the electron transfer step is increased, leading to diminished electrochemical activity.

However, it is essential to note that the reduction in the oxidation–reduction potential of $FeCl_2$ comes at the cost of a decreased open-circuit voltage that the electrolyte can

provide. When using the commonly used $VCl_3$ as the negative electrode electrolyte, the cell can only deliver an open-circuit voltage of 0.74 V during discharge, significantly limiting the further improvement of SFBs performance. For TEMPO and 4-OH-TEMPO, which can offer higher voltages, further attempts were made to introduce magnetic nanoparticles to optimize the solution's oxidation–reduction capability, enhancing the semiconductor's photocurrent density in the solution.

### 3.2. Impact of Magnetic $Fe_3O_4$ Nanoparticles on Redox Couples and Photoelectrochemistry

The preparation of $Fe_3O_4$ nanoparticles through co-precipitation is a commonly employed method. In this study, the $Fe_3O_4$ nanoparticles synthesized using this technique were subjected to X-ray diffraction (XRD) analysis, as depicted in Figure 4a. A comparison with standard reference patterns revealed characteristic diffraction peaks indicative of magnetic $Fe_3O_4$. Simultaneously, the morphology and size of the prepared nanoparticles were observed using scanning electron microscopy. As shown in Figure 4b, $Fe_3O_4$ nanoparticles predominantly exhibit irregular spherical shapes with relatively uniform sizes. Despite some particle aggregation during measurement, it is evident that they are composed of numerous particles around 30 nm in size. Figure 4c,d show X-ray energy spectrum analysis (EDS), from which the distribution of Fe and O elements can be observed. Quantitative analysis indicates atomic percentages of 37.1% for Fe and 62.9% for O.

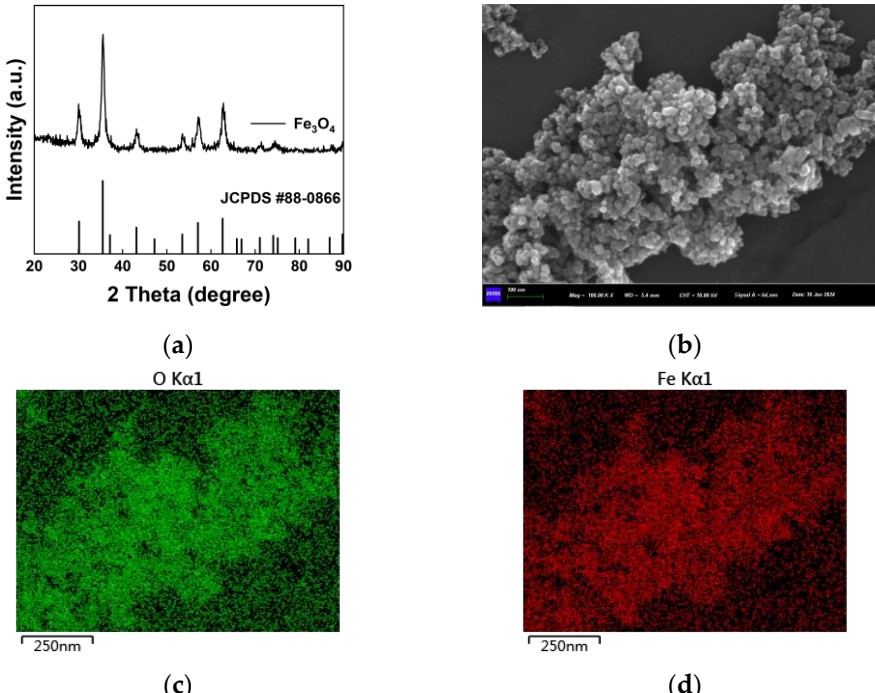

**Figure 4.** (**a**) XRD; (**b**) SEM of synthesized $Fe_3O_4$ nanoparticles; the EDS of $Fe_3O_4$ nanoparticles: (**c**) O; (**d**) Fe.

Furthermore, from Figure 5, it can be observed that the initial viscosity and initial conductivity of the various electrolytes are comparable, each approximately 50 mPa·s and 2 mS·cm$^{-1}$, respectively. With the increase in the mass ratio of nanoparticles, both the electrolyte viscosity and conductivity show an upward trend. When the mass ratio is 0.05 wt%, the viscosity of 4-OH-TEMPO exhibits the maximum increase, approximately 10%, whereas its conductivity experiences the smallest increment, only 8.2%. The impact of nanoparticles on the form of redox couples was investigated through UV-Vis curve analysis of the test solution [25]. As depicted in Figure 6, the original electrolytes of TEMPO and 4-OH-TEMPO exhibit two characteristic absorption peaks at 202 nm and 242 nm, respectively [37,38]. The $FeCl_2$ electrolyte shows two distinct absorption peaks at 210 nm

and 305 nm [39]. Adding a certain amount of $Fe_3O_4$ to the original electrolyte does not cause any significant shift in the absorption peak positions, indicating the absence of chemical interactions between the redox couples in the electrolyte and the nanoparticles. Meanwhile, due to the light absorption characteristics of $Fe_3O_4$ particles in the range of 200–450 nm [40], the absorbance of the nanofluid electrolyte increases in this wavelength range.

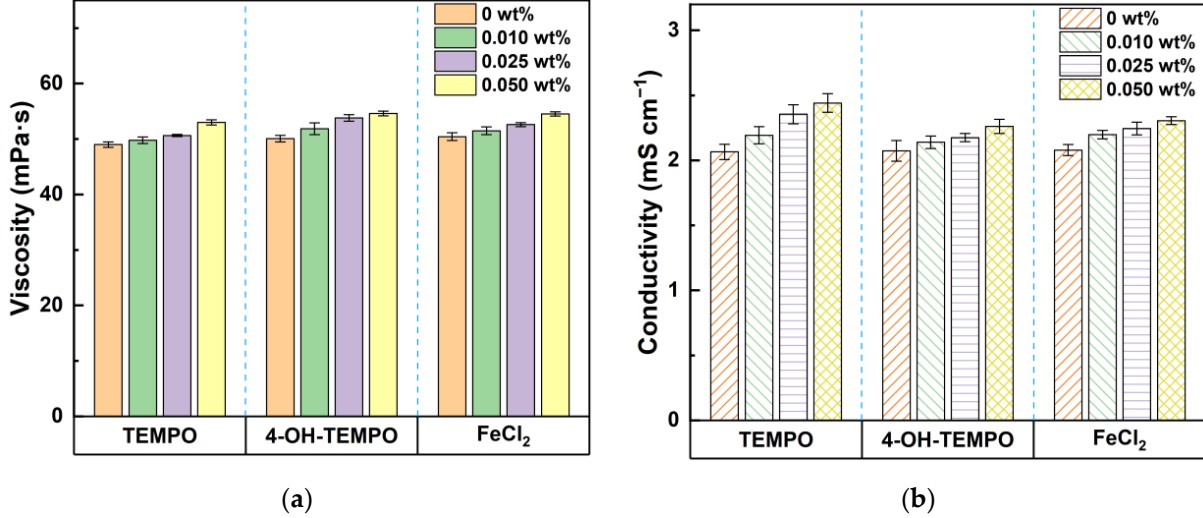

**Figure 5.** (**a**) Viscosity; (**b**) conductivity of the original electrolyte and those with the addition of nanoparticles with different mass ratios.

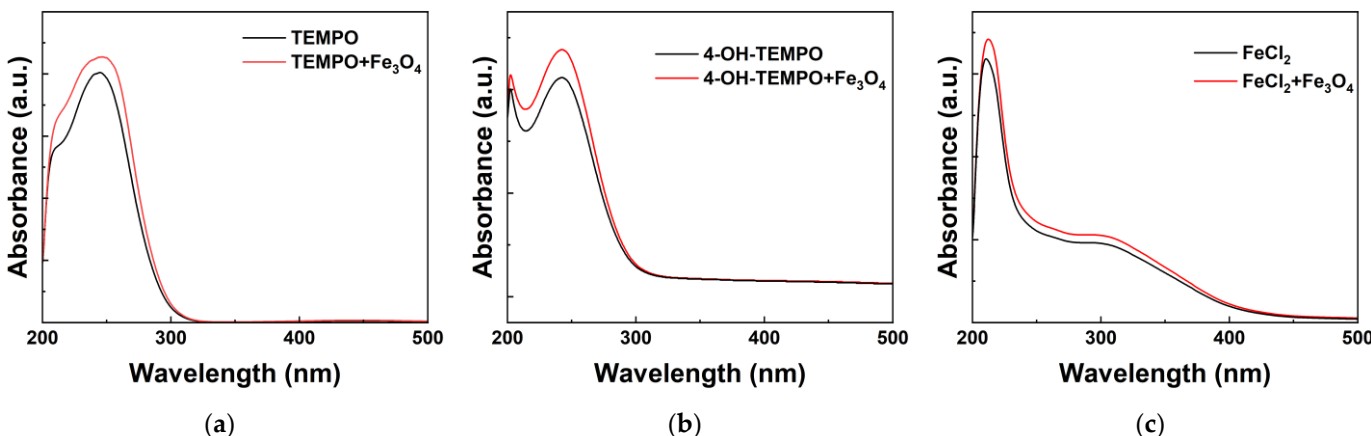

**Figure 6.** UV-visible spectra of the original electrolyte and the magnetic fluid electrolyte with the optimal mass ratio for (**a**) TEMPO, (**b**) 4-OH-TEMPO, and (**c**) $FeCl_2$.

As shown in Figure 7, the oxidation and reduction peak current densities of the three redox pairs increase compared to the original electrolyte upon adding $Fe_3O_4$ nanoparticles. The most significant improvement is observed in TEMPO and 4-OH-TEMPO electrolytes with a mass ratio of 0.05 wt%. In these cases, the oxidation peak current densities increase by 15.7% and 16%, while the reduction peak current densities increase by 13% and 22%, respectively. Notably, beyond a magnetic $Fe_3O_4$ mass ratio of 0.025 wt%, the peak current densities in TEMPO and 4-OH-TEMPO electrolytes show diminishing increases with further mass ratio increments. Therefore, 0.025 wt% is considered the optimal mass ratio. For the $FeCl_2$ electrolyte, the optimal $Fe_3O_4$ mass ratio is 0.01 wt%, resulting in a 14% increase in the oxidation peak and a 10% increase in the reduction peak. Additionally, as the mass ratio increases, the enhancement of the redox kinetics of the electrolyte becomes less pronounced, and the peak reduction current even slightly decreases. On the other hand, the more significant addition amount leads to issues such as particle aggregation and

excessive viscosity, which are not conducive to practical battery application. By analyzing and fitting the EIS graphs, as shown in Table 2, the introduction of nanoparticles reduces the internal resistance, with the most significant decrease observed in the solution's Ohmic resistance and charge transfer resistance in the TEMPO solution. This is attributed to the increased liquid conductivity.

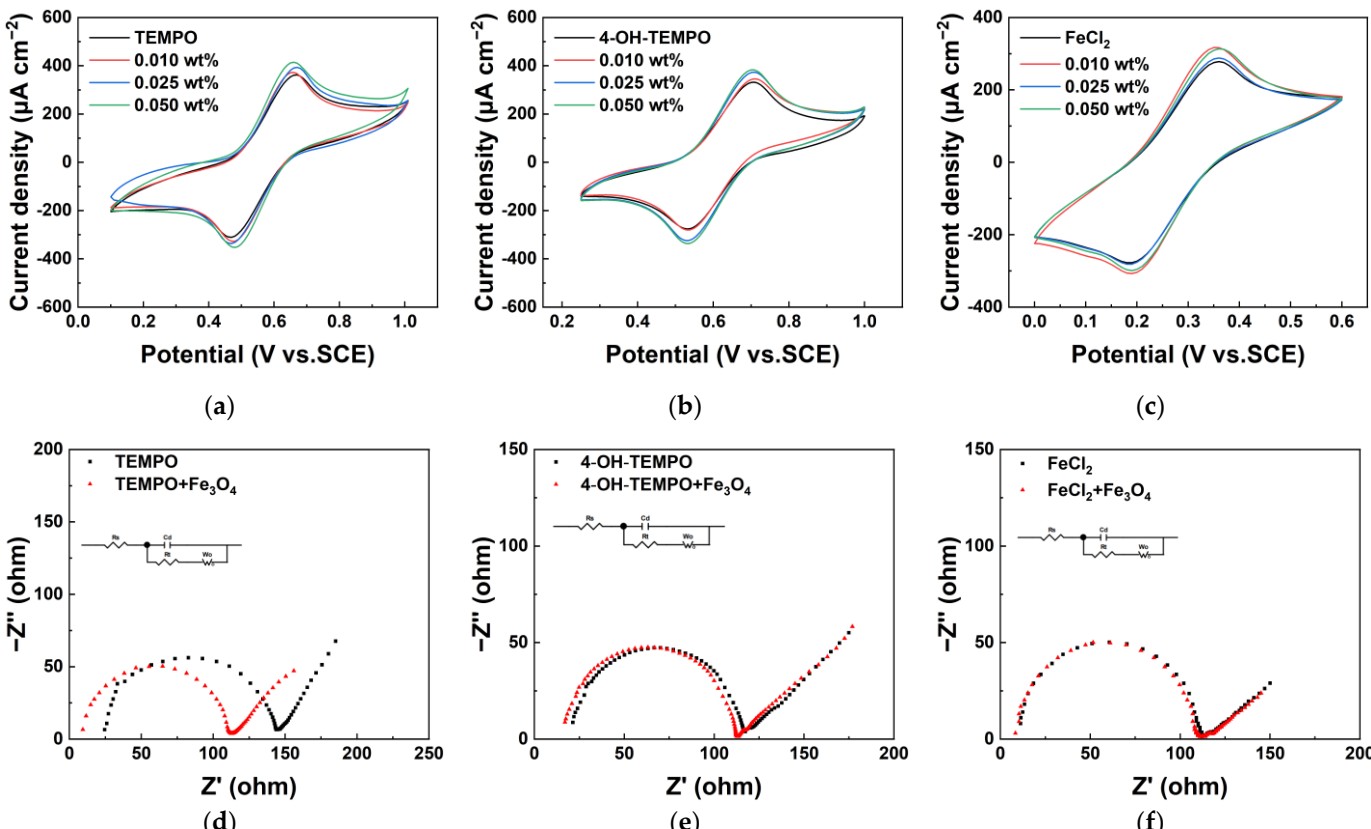

**Figure 7.** CV curves of the original electrolyte and the magnetic fluid electrolyte for (**a**) TEMPO; (**b**) 4-OH-TEMPO; (**c**) $FeCl_2$ at the scan rate of 10 mV s$^{-1}$; EIS curves of the original electrolyte and the magnetic fluid electrolyte with the optimal mass ratio for (**d**) TEMPO; (**e**) 4-OH-TEMPO; (**f**) $FeCl_2$.

**Table 2.** The parameters resulting from fitting the impedance plots with the equivalent circuit model are in Figure 6b,e,f.

|  | Rs/Ω | Rt/Ω | Wo/Ω |
|---|---|---|---|
| TEMPO+$Fe_3O_4$ | 9.22 | 101.2 | 189.3 |
| 4-OH-TEMPO+$Fe_3O_4$ | 16.33 | 95.13 | 215.6 |
| $FeCl_2$+$Fe_3O_4$ | 8.62 | 100.1 | 189.4 |

Finally, we tested the instantaneous photocurrent curve of the three magnetic nanofluid electrolytes to compare and assess the impact of magnetic $Fe_3O_4$ nanoparticles on the photoelectrochemical performance of different redox couples. As illustrated in Figure 8, adding an appropriate amount of magnetic $Fe_3O_4$ nanoparticles in the solution can optimize the physicochemical properties of the electrolyte, effectively enhancing the photoelectrode's photocurrent density. The photocurrent densities of the three magnetic nanofluid electrolytes increased to 25.0 μA cm$^{-2}$, 44.7 μA cm$^{-2}$, and 79.7 μA cm$^{-2}$, with respective enhancements of 36.6%, 17.0%, and 7.5%.

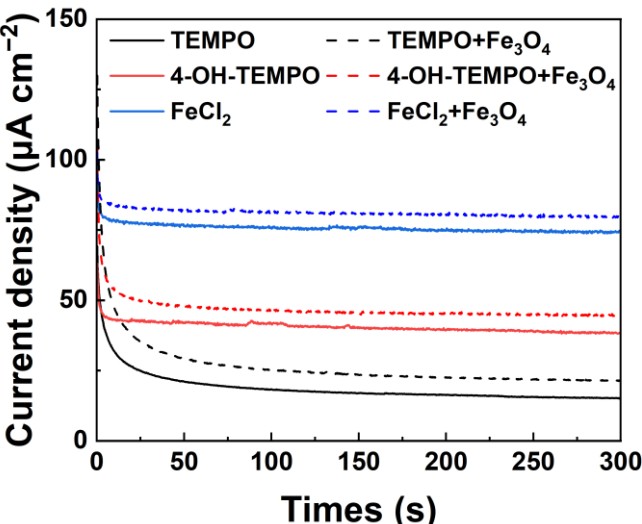

**Figure 8.** I-t curve before and after adding magnetic nanoparticles at the optimal mass ratio in different redox couples.

In summary, magnetic $Fe_3O_4$ nanoparticles were successfully synthesized through a co-deposition method, and nanofluid electrolytes were prepared by incorporating these nanoparticles into the electrolyte with varying mass ratios. The introduction of nanoparticles enhanced the conductivity, peak current density, and reversibility of both TEMPO and 4-OH-TEMPO electrolytes while concurrently reducing their ohmic resistance. Furthermore, the excellent conductivity of $Fe_3O_4$ nanoparticles facilitated the creation of an electron network in the solution, expediting ion migration. Consequently, the photoelectrochemical performance of both nanofluid electrolytes was improved, leading to a significant enhancement in the photocurrent density of the photoelectrode in nanofluid electrolytes. This improvement not only resulted in higher photovoltage for solar flow batteries but also generated more optimal photocurrent, thereby enhancing the overall usability of the battery system.

## 4. Conclusions

In conclusion, the photoelectrode exhibits varying flat-band potentials and carrier concentrations in different electrolytes, resulting in distinct photochemical reaction capabilities. When the photoelectrode is immersed in the $FeCl_2$ solution with lower oxidation-reduction potential and internal resistance, it achieves the maximum photocurrent density due to the formation of a lower flat-band potential and higher carrier concentration. However, this comes at the expense of the inability to generate a higher open-circuit voltage. This study further attempts to enhance the photoelectrode's photochemical reaction capability in the solution by preparing a magnetic nanofluid. Specifically, for the TEMPO and 4-OH-TEMPO electrolytes, which can provide higher open-circuit voltage, the photocurrent density generated by the photoanode with magnetic nanofluid electrolytes increased by 36.6% and 17.0%, respectively. The utilization of magnetic nanoparticles in the SFBs holds significant research potential, offering a novel pathway to enhance battery efficiency and propel SFBs toward commercialization.

**Author Contributions:** Conceptualization, H.S. and Q.X.; Methodology, Z.G.; Validation, H.S. and P.L.; Resources, Z.Z. and Q.X.; Software, Q.M.; Writing—original draft preparation, Z.G., Z.Z. and P.L.; Writing—review and editing, Z.G. and Q.X.; Funding acquisition, Q.M. and Q.X. All authors have read and agreed to the published version of the manuscript.

**Funding:** NSFC: China (No. 52276066, No. 51676092); Six-Talent-Peaks Project in Jiangsu Province (No. 2016-XNY-015); High-Tech Research Key Laboratory of Zhenjiang City (No. SS2018002).

**Institutional Review Board Statement:** Not applicable.

**Informed Consent Statement:** Not applicable.

**Data Availability Statement:** Data are contained within the article.

**Acknowledgments:** The supports (including materials used for experiments and helpful discussions) from Xiaozhong Shen and Lu Lu (both from Wuxi Vocational Institute of Commerce) are gratefully acknowledged.

**Conflicts of Interest:** The authors declare no conflicts of interest.

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
