# Peer review of "Optimal Selection for Redox Couples and Enhanced Performance through Magnetic Nanofluid Electrolyte in Solar Flow Batteries"

_magnetochemistry, doi:10.3390/magnetochemistry10020011_

Round 1

Reviewer 1 Report

Comments and Suggestions for Authors

1. The article does not contain information about the colloidal stability of electrolytes with the addition of nanoparticles. The article makes no mention of any surfactants that were coated on the nanoparticles. In this case, they should settle to the bottom.

2. None of the experimental graphs indicate the measurement error

3. What is the repeatability of the results. Do the curves coincide in subsequent experiments?

4. There is no justification for choosing exactly these concentrations of nanoparticles

5. From graph 7 it is clear that the curves for pure electrolytes have reached saturation, and for electrolytes with the addition of nanoparticles they continue to decline. Why was this particular time interval of the experiment? Will the advantage of electrolytes disappear with the addition of nanoparticles over a time period of 200 seconds.

Reviewer 2 Report

Comments and Suggestions for Authors

The manuscript reported and compared the chemical and photoelectrochemical properties of three commonly used redox couples for SFB applications. Magnetic Fe3O4 nanoparticles were introduced to optimize the electrolyte. It is found that FeCl2/ FeCl3 redox coupled with TiO2 photoelectrodes exhibits the highest photoelectric current density. Moreover, for TEMPO and 4-OH-TEMPO electrolytes with a higher OCV, the electrochemical activity can be enhanced and the solution ohmic resistance is reduced by introducing magnetic nanoparticles and forming the magnetic nanofluid. In this way, the photocurrent density increases by 40.3% and 15.5%, respectively.

I consider the content of this manuscript will definitely meet the reading interests of the readers of the Magnetochemistry journal. However, the discussion and explanation should be further improved. I suggest giving a minor revision and the authors need to clarify some issues or supply some more experimental data to enrich the content. This could be comprehensive and meaningful work after revision.

Detailed comments can be found in the PDF file.

Reviewer 3 Report

Comments and Suggestions for Authors

This manuscript entitled "Optimal Selection for Redox Couples and Enhanced Performance through Magnetic Nanofluid Electrolyte in Solar Flow Batteries" discusses about compares the chemical and photoelectrochemical properties of three commonly used redox couples. The effect of the magnetic properties of the electrolytes seems to be the main key point of this study. This manuscript is suggested for acceptance after considering the following points (major revision):

Abstract: Please state the importance of this work in one sentence at the end of the abstract. Notably, the importance has been well defined in the last paragraph of the Introduction section.

Line 51-67: In explaining SFB, are there any reference(s) existing? Please cite it (them) if any.

 Line 84-88: The referee suggests merging these lines with the last paragraph so that it would be the only paragraph focusing on the goals of this work.

Methods: In Figure 4, XRD measurement is shown. Please state the measurement details in the method section, including the instrument, X-ray wavelength, etc.

Morphology: Showing SEM images of the nanoparticles is important to confirm the existence of the nanoparticles.

Structural properties: Supporting the XRD spectra, is there any additional measurement, such as EDX or EDS, to confirm the elemental contents of the nanoparticles?

This manuscript separates the "Results" and "Discussion" sections. What if both sections are merged, while the last paragraph is placed under a new section "Conclusions"? Please consider this since the "Discussion" section is quite short for being separated from the results.

Conclusion: Please state the importance of this work in one sentence at the end of the conclusion.

Round 2

Reviewer 3 Report

Comments and Suggestions for Authors

The authors have addressed the comments very well. The referee suggests accepting the manuscript in the revised form.